# Cell softness regulates tumorigenicity and stemness of cancer cells

Jiadi Lv[1,†], Yaoping Liu[2,†], Feiran Cheng[1,†] (ID), Jiping Li[3], Yabo Zhou[1], Tianzhen Zhang[1], Nannan Zhou[1], Cong Li[1], Zhenfeng Wang[1], Longfei Ma[1], Mengyu Liu[1], Qiang Zhu[1], Xiaohan Liu[1], Ke Tang[4], Jingwei Ma[4], Huafeng Zhang[4], Jing Xie[1], Yi Fang[5], Haizeng Zhang[5], Ning Wang[6], Yuying Liu[1,7,**] (ID) & Bo Huang[1,4,7,*] (ID)

## Abstract

Identifying and sorting highly tumorigenic and metastatic tumor cells from a heterogeneous cell population is a daunting challenge. Here, we show that microfluidic devices can be used to sort marker-based heterogeneous cancer stem cells (CSC) into mechanically stiff and soft subpopulations. The isolated soft tumor cells (< 400 Pa) but not the stiff ones (> 700 Pa) can form a tumor in immunocompetent mice with 100 cells per inoculation. Notably, only the soft, but not the stiff cells, isolated from CD133[+], ALDH[+], or side population CSCs, are able to form a tumor with only 100 cells in NOD-SCID or immunocompetent mice. The Wnt signaling protein BCL9L is upregulated in soft tumor cells and regulates their stemness and tumorigenicity. Clinically, BCL9L expression is correlated with a worse prognosis. Our findings suggest that the intrinsic softness is a unique marker of highly tumorigenic and metastatic tumor cells.

**Keywords** BCL9L; metastasis; microfluidic sorting; soft tumor cells; stemness
**Subject Categories** Cancer; Cell Adhesion, Polarity & Cytoskeleton; Stem Cells & Regenerative Medicine
**The EMBO Journal (2021) 40: e106123**

## Introduction

The notion of "cancer stem cells" or tumorigenic cells has been based on the observation that only a very small population of cells from a tumor can seed and form a tumor in severe combined immunodeficient (SCID) mice (Lapidot *et al*, 1994; Al-Hajj *et al*, 2003; Hope *et al*, 2004; Singh *et al*, 2004; O'Brien *et al*, 2007; Quintana *et al*, 2008; Schatton *et al*, 2008). These tumorigenic cells are viewed as the source of treatment resistance and relapse of a tumor, making them a tempting therapeutic target. However, despite intensive studies, the properties of this crucial tumor cell subset remain poorly understood. Furthermore, rigorous methods are not available to isolate these cells from a tumor, because the conventional cell surface markers are unreliable and highly variable among different cancers (Hope *et al*, 2004; Dieter Sebastian *et al*, 2011). Thus, developing a method that can effectively sort and define tumorigenic cells is extremely desirable. Published reports have highlighted the importance of the mechanical properties of a living cell in cell behaviors and functions (Engler *et al*, 2006; Chowdhury *et al*, 2010; Urbanska *et al*, 2017). It is known that cells apply actomyosin-dependent contractile forces in response to the increasing stiffness of the extracellular matrices (ECMs) (Discher *et al*, 2005; Irianto *et al*, 2016). Such endogenous contraction, in turn, can elevate cell stiffness (Wang *et al*, 1993). Moreover, to properly sense and respond to the surrounding mechanical cues, the stiffness of a cell should match that of the ECMs (Discher *et al*, 2005; Discher *et al*, 2009; Wu *et al*, 2018), suggesting that soft cells should survive in a soft stroma and stiff cells behave optimally in a stiff niche. As a support, we have demonstrated that 3D soft fibrin matrices promote H3K9 demethylation and increase Sox2 expression and self-renewal of melanoma stem cells, whereas stiff ones exert opposite effects (Tan *et al*, 2014). In line with this notion, cells with various degrees of stiffness can co-exist within the same tumor tissue, due to the heterogeneity of the tumor mechanical microenvironments (Plodinec *et al*, 2012; Elosegui-Artola *et al*, 2014). Despite such understanding, direct evidence that the cellular softness functions as

1   Department of Immunology & National Key Laboratory of Medical Molecular Biology, Institute of Basic Medical Sciences, Chinese Academy of Medical Sciences (CAMS) & Peking Union Medical College, Beijing, China
2   Institute of Microelectronics, Peking University, Beijing, China
3   Beijing Smartchip Microelectronics Technology Company Limited, Beijing, China
4   Department of Biochemistry & Molecular Biology, Tongji Medical College, Huazhong University of Science & Technology, Wuhan, China
5   National Cancer Center/Cancer Hospital, CAMS, Beijing, China
6   Deaprtment of Mechanical Science and Technology, The Grainger College of Engineering, University of Illinois at Urbana-Champaign, Urbana, IL, USA
7   Clinical Immunology Center, CAMS, Beijing, China
    *Corresponding author. Tel: +86 10 69156447; E-mail: 13161773902@163.com
    **Corresponding author. Tel: +86 10 69156464; E-mail: tjhuangbo@hotmail.com
    †These authors contributed equally to this work

a basic feature for tumorigenic cells remains elusive. In this study, we develop a method to separate soft cells from stiff ones and provide evidence that these soft cells are highly tumorigenic and possess the ability to metastasize.

## Results

### Soft tumor cells are sorted by microfluidic chip

Indeed, atomic force microscopy (AFM) analysis showed that the stiffness of tumor cells in mouse breast cancer (4T1), human breast cancer (MCF-7), mouse B16 melanoma, and primary human melanoma (MP-1) was highly variable, ranging from 0.2 to 1.3 kPa. Notably, more than 60% of tumor cells had at least a stiffness of 0.7 kPa and less than 10% tumor cells had the stiffness below 0.4 kPa (Fig EV1A). Since the softness (the inverse of stiffness) renders a cell a greater degree of deformability, we thus rationally explored this as a way to separate soft tumor cells from stiff ones (Mohamed *et al*, 2009; Zhang *et al*, 2012).

Here, we designed a unique microfluidic chip for label-free cell sorting based on the cells' physical characteristics (e.g. stiffness/deformability). The proposed microfluidic device consists of two major components, flow channels, and microweir structures (Fig 1A, upper). The gap between the microweir and main channel can act as a passive and selective barrier to isolate cells with a variety of stiffness (Fig 1A, lower). Considering that tumor cells usually

have the size around 20–25 μm (Hosokawa *et al*, 2010; Hvichia *et al*, 2016), in this study, the microfluidic chips were fabricated with a 15 μm gap generation by setting the height of the microweir and flow channels at 25 and 40 μm, respectively. According to previous literatures (Mohamed *et al*, 2009; Zhang *et al*, 2012), four different types of chips (with different lengths, widths of microweir structure, spaces between microweir structures, and total numbers of microweir structures, shown in Fig EV1B and Table EV1) were designed and tested to perform the sorting. Cells (density ranging from $1 \times 10^4$ to $2 \times 10^4$/ml) with different degree of stiffness were injected into the chip via a syringe pump at a flow flux of 10 μl/min (Fig 1B). The cells, as they flowed out from the outlet, were collected as soft cells. In addition, we also pumped the bulk tumor cells through the microfluidic channels with a larger gap (18 μm), and inversely washed the channels. Cells then flowed out from backflow inlet and were collected as stiff cells (Fig 1B). To verify that these separated cells are authentically soft or stiff ones, we used AFM to measure the stiffness of cells. Indeed, the cells which flowed out from the outlet of each type of microtube were much softer than the cells which flowed out from the inlet (Fig 1C). Notably, the microfluidic tube #2 (260 μm distance between ridges and 11.44 mm channel length) displayed the highest sorting efficiency (Fig EV1C) and thus was used for the following experiments. Tumor cells isolated from melanoma (B16F1 and MP-1) and breast cancer (4T1 and MCF-7) by the microfluidic tube all displayed the soft trait (Fig 1D). We found that the separated stiff and soft cells had a similar cellular size (Fig EV1D). Also, the stiff and soft tumor cells

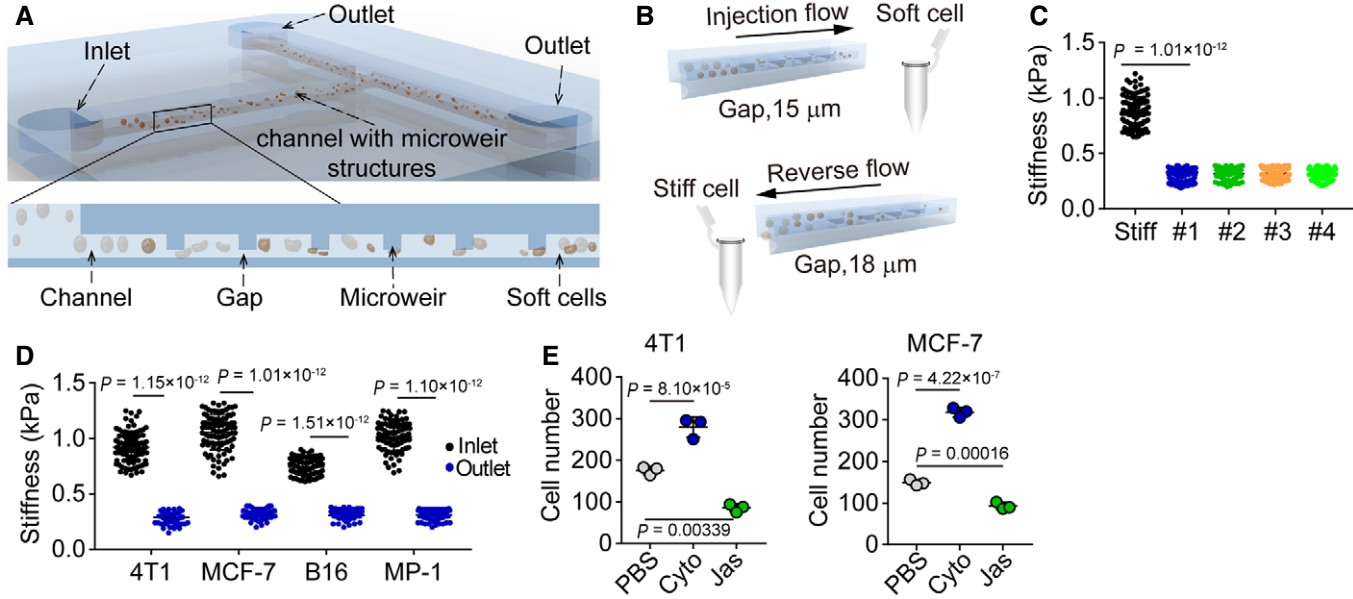

**Figure 1. Soft tumor cells are sorted by microfluidic chip.**

A Schematic of microfluidic chip.

B Schematic of working principle for cell screening based on stiffness differences.

C The screening efficiency of four different types of chip was detected in 4T1 cells. *n* = 100.

D The stiffness of cells screened from the outlet or backflow inlet was measured by AFM. *n* = 100.

E The number of soft 4T1 or MCF-7 cells from the outlet was calculated after treatment with or without Cyto (5 μM) or Jas (50 nM) for 4 h or 12 h, respectively.

Data information: Mann–Whitney test (D), Kruskal–Wallis test (C) or one-way ANOVA (E). The data represent mean ± SD of three independent experiments.

displayed a similar growth curve in *in vitro* culture (Fig EV1E). Given that the stiffness of the isolated soft tumor cells was less than 0.4 kPa and the stiffness of the stiff cells was larger than 0.65 kPa, we defined soft cells as those with < 0.4 kPa stiffness and stiff cells as those with > 0.7 kPa stiffness in this study. In addition, we mixed the isolated soft and stiff cells for re-separation using the microfluidic chip. We found that the distribution of soft cells before re-separation was completely consistent with those following re-separation (Fig EV1F), indicating that this microfluidic tube does not alter the original stiffness of the cells. F-actin is an essential element that contributes to cellular stiffness (Wang *et al*, 1993). Cytochalasin D (Cyto), an inhibitor of actin polymerization, can decrease cell stiffness; In contrast, jasplakinolide (Jas), a natural cyclodepsipeptide that is a potent inducer of actin polymerization, can increase cell stiffness (Fig EV1G). Following the Cyto or Jas treatment of tumor cells (4T1, MCF-7, B16, or MP-1), the number of soft tumor cells from the outlet was increased by Cyto but decreased by Jas (Figs 1E and EV1H). Thus, a marginal population of tumor cells with the mechanical property of softness can be separated from the bulk cells using the microfluidic chip.

## Soft cells are highly tumorigenic and metastatic

Next, we investigated the biological property of the soft tumor cells. Our previous studies had demonstrated that tumorigenic cells rather than differentiated tumor cells are selected and grow in 90 Pa soft 3D fibrin gels (Liu *et al*, 2012; Liu *et al*, 2018); and these selected cells are physically much softer than the differentiated counterparts and can be represented by CD133$^{hi}$ melanoma cells or ALDH$^+$ breast cancer cells *in vivo* (Liu *et al*, 2018). These findings prompted us to hypothesize that mechanical softness might be a common feature for tumorigenic cells. To test this hypothesis, we seeded the soft or stiff tumor cells following separation by a microfluidic channel (B16, MP-1, 4T1, MCF-7) into the 90 Pa soft 3D fibrin gels. We found that greater than 95% of soft tumor cells could form colonies, while stiff tumor cells only formed few colonies with a much smaller size (Figs 2A and EV2A). Meanwhile, we found that the softness of the isolated soft MCF-7 cells could be kept in soft fibrin gels, while, the soft cells, if seeded in rigid culture plates for 4 h, started to become stiff and reached the stiff peak 12 h later. Next, we injected 100 separated soft or stiff cells (4T1 and MCF-7) into the mammary fat pads of NOD/SCID IL-2Rγ-null (NSG) mice. Twelve weeks later, the 100 soft cells could form a tumor *in situ* with a relatively high frequency (6/10 for 4T1, 4/10 for MCF-7), while the 100 stiff cells injected did not form a tumor (Fig 2B). Moreover, 100 soft 4T1 cells could even form a tumor in immunocompetent mice (wild-type BALB/c) with a rate of tumor formation (3/8), suggesting that the soft tumor cells have a highly tumorigenic ability. In addition, when we performed limiting dilution experiments to quantify the frequency of tumors for each condition of soft or stiff cell inoculation (O'Brien *et al*, 2007), we found that the highest frequency of tumor formation occurred in the soft cell group (Fig 2C). Moreover, by conducting serial transplants with 100 soft tumor cells, we found that the tumor size and appearance of the first implantation was similar to the second and third generation of passaged tumors in the mice (Figs 2D and EV2D). A cardinal feature of malignant melanoma is its inclination to metastasize to the lungs. Eight weeks following an intravenous injection of 100 soft or stiff cells separated

from B16-F1 or MP-1 in NSG mice, metastatic tumors in the lungs were visible from the soft cell group. Interestingly, even as few as 10 soft cells could generate metastatic tumors (2/12 for B16 or 2/8 for MP-1), but no metastatic tumors were detected in the lungs following the injection of 100 stiff cells (Fig 2E and F). Then, we used the 4T1 cell lung metastasis model to further validate this result. The soft or stiff 4T1 cells were injected into the mammary fat pads of WT BALB/c mice. Eight weeks later, the mice were sacrificed for H&E staining of the lungs, showing metastatic tumors (4/8) in the soft cell group (Fig EV2E and F). However, no lung metastatic tumors (0/8) were observed in the stiff cell group. In line with these *in vivo* results, the soft tumor cells displayed greater ability to migrate and invade *in vitro*, compared with the stiff cells (Fig EV2G–I). Together, these data suggest that soft tumor cells are highly tumorigenic in their ability to form a tumor at both primary and metastatic sites.

## Softness is a physical marker for tumorigenic cells

Next, we asked whether this mechanical softness could function as a useful marker for tumorigenic cells. Chemical molecules on the cell surface or in the cytosol, such as the enzyme ALDH1, have been widely used to denote cancer stem cells despite their unreliability and low specificity (Ginestier *et al*, 2007; Douville *et al*, 2009). When we used softness to analyze ALDH1$^+$ and ALDH1$^-$ breast cancer cells (4T1 and MCF-7), we found that around 65% ALDH1$^+$ cells were soft and around 6% ALDH1$^-$ cells were soft (Fig 3A). Intriguingly, inoculation of 100 soft either ALDH1$^+$ or ALDH1$^-$ tumor cells were able to form a tumor in either NSG or WT mice; however, inoculation of 100 stiff ALDH1$^+$ or ALDH1$^-$ tumor cells could not form a tumor, even in NSG mice (Figs 3B and EV3A). In addition, the inoculation of 100 unseparated ALDH1$^+$ tumor cells displayed half the tumorigenic capability of the sorted soft ones; however, these sorted soft ones had the similar tumorigenic capability as the sorted ALDH1$^+$ soft tumor cells (Figs 3B and EV3B). Consistently, only soft ALDH1$^-$ (100) but not stiff ALDH1$^+$ ones were able to form lung metastatic micronodules (Fig 3C and D). Besides 4T1 and MCF-7, similar results were also obtained from B16 and MP-1 melanoma cells. We found that ~ 60% CD133$^+$ and ~ 4% CD133$^-$ cells were soft (Fig EV3C), and 100 CD133$^-$ soft cells had the ability to form a subcutaneous tumor and lung metastasis, but this was not observed with the 100 CD133$^+$ stiff cells (Fig EV3D and E). Thus, surface marker-defined cancer stem cells may contain both the soft and stiff subpopulations and only the soft subset has a tumorigenic capability. To further validate this conclusion, we used a comparable approach to isolate a stem cell-like side population (SP) from tumor cells, based on the efflux of Hoechst 33342 (Golebiewska *et al*, 2011). Intriguingly, among SP cells, around 50–60% cells displayed a soft property (Figs 3E and EV3F). Also, the isolated SP and non-SP (NSP) cells (4T1 or MCF-7) were further separated into stiff and soft subpopulations using microfluidic devices. As expected, the injection of 100 soft NSP tumor cells into the mammary fat pads of NSG mice led to visible tumor formation within 12 weeks at a ~ 50% frequency, but the injection of stiff SP$^+$ cells did not cause detectable tumors (Fig 3F). Similar results were obtained from the stiff SP or soft NSP B16 or MP-1 cells (Fig 3F). In addition, metastatic micronodules in the lungs were found by H&E staining from the soft NSP group but not from the stiff SP group on day 80 (Figs 3G and H, and EV3FG and H). Thus, SP cells also contain soft subpopulation

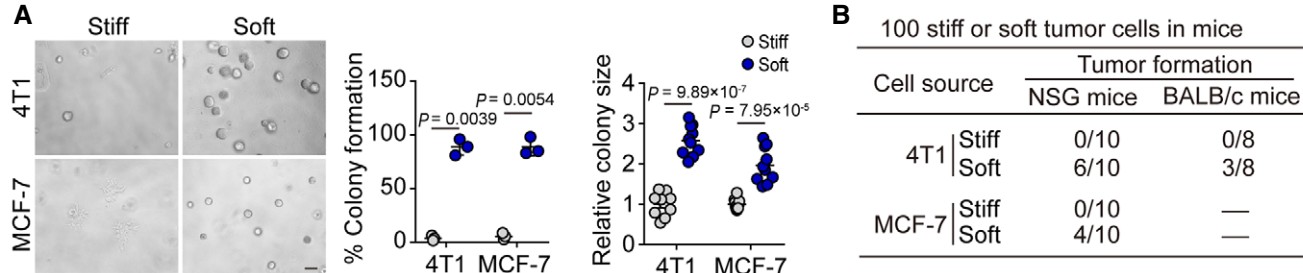

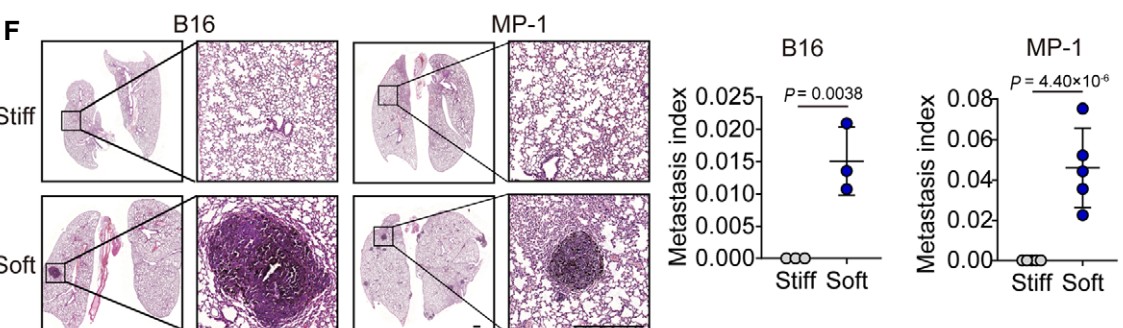

**Figure 2. Soft tumor cells have the ability to form tumors in mice.**

A   Soft or stiff 4T1 or MCF-7 cells were isolated by the microfluidic chip, and then seeded (500 cells) in 90 Pa soft 3D fibrin gel for 3 days. The percentage of colonies formed was calculated and the colony size was recorded. Scale bar, 100 μm.

B   The soft or stiff 4T1 or MCF-7 cells (100 cells) were injected into the mammary fat pads of NSG or BALB/c mice. Twelve weeks later, the tumor formation was recorded. *n* = 8–10.

C   The same as (B), except that different number of 4T1 cells were injected into the BALB/c mice. *n* = 10.

D   The tumor-forming capacity from primary xenografts (F1) and tumors passaged into secondary (F2) and tertiary (F3) recipients induced by injecting 100 soft 4T1 or B16 cells into the BALB/c or C57BL/6 mice. *n* = 10.

E   Stiff or soft B16 or MP-1 cells (100 or 10 cells) were injected into the NSG mice by tail veil. *n* = 8–12.

F   NSG mice were intravenously injected with 100 soft or stiff B16 or MP-1 cells. Eight weeks later, the lung metastasis was analyzed by H&E staining. The metastasis index was defined as the percentage of total metastatic nodule area to the total lung area based on the calculation from 10 slides. Scale bar, 0.5 mm. *n* = 3 (for B16) or 5 (for MP-1) mice with metastatic tumor.

Data information: Two-tailed paired Student's *t*-test (A and F). The data represent mean ± SD. *n* = 3 independent experiments in (A).

and employ them to grow a tumor at the primary and metastatic sites. Together, these results suggest that intrinsic softness may be a suitable characteristic to mark tumorigenic cells.

## Soft tumor cells upregulated Wnt-BCL9L pathway for stemness development

Furthermore, we examined the biological difference between soft and stiff cells at the genetic levels in order to better understand the merit of softness as a marker for tumorigenic cells. To this end, we began by sequencing the entire genome of the soft tumorigenic, ALDH[+], and stiff differentiated cells, respectively. We did not find a difference at the DNA levels among the three groups. Subsequently, we assessed the gene expression pattern by mRNA sequencing. Using principal component analysis (PCA), we found that the difference between the soft and stiff groups was much larger than that between the CSC and stiff groups, while the internal variation in the soft group was smaller than that in the CSC group (Fig 4A). Meanwhile, the differentially expressed genes among the soft, stiff, and CSC cells were also shown by

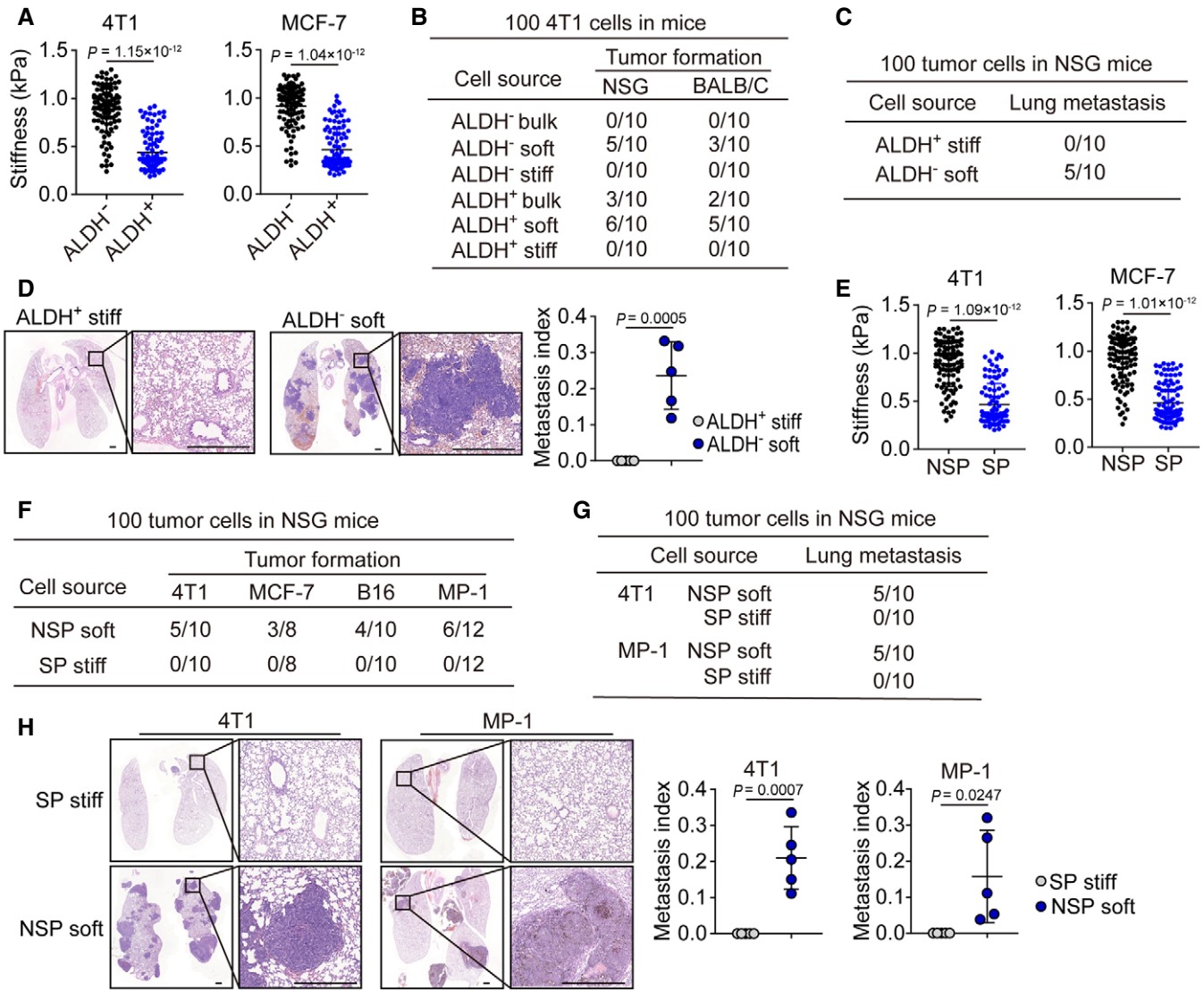

**Figure 3. Tumorigenic cells are characterized by mechanical softness.**

A   The cellular stiffness of ALDH$^-$ and ALDH$^+$ 4T1 or MCF-7 cells sorted by flow cytometry were determined by AFM. $n = 100$.

B   The stiff or soft cells were isolated from ALDH$^-$ or ALDH$^+$ 4T1 cells by microfluidic chip. Then, these cells were injected into the mammary fat pads of NSG or BALB/c mice (100 cells/mouse). Twelve weeks later, the tumor formation was recorded. $n = 10$.

C, D   The 100 stiff ALDH$^+$ or soft ALDH$^-$ 4T1 cells were injected into the mammary fat pads of NSG mice. Eight weeks later, the lung sections were H&E stained. The lung metastatic mice were counted (C), and the metastasis index was calculated (D). Scale bar, 0.5 mm. $n = 5$ mice with metastatic tumor.

E   The side population (SP) or non-SP (NSP) from 4T1 or MCF-7 cells were sorted by flow cytometry, and then, the cellular stiffness of those cells was measured by AFM. $n = 100$.

F   The 100 soft NSP or stiff SP- 4T1, MCF-7, B16, or MP-1 cells were injected into the mammary fat pads (4T1 and MCF-7) or subcutaneous tissue (B16 and MP-1) of NSG mice. The tumor formation in the lungs was recorded.

G, H   The 100 soft NSP or stiff SP- 4T1 or MP-1 cells were injected into the mammary fat pads (for 4T1) of NSG mice or by tail vein into NSG mice (for MP-1). The tumor formation in the lungs was recorded (G). Six to eight weeks later, mice were sacrificed and the lung sections were H&E stained. The metastatic micronodules in the lung were counted, and the metastasis index was calculated (H). Scale bar, 0.5 mm. $n = 5$ mice with metastatic tumor.

Data information: Mann–Whitney test (A and E), or two-tailed paired Student's $t$-test (D and H). The data represent mean $\pm$ SD.

Venn diagram (Fig 4B). Gene ontology analysis identified these differentially expressed gene families to be associated with stem cell proliferation, cell migration, and immune system processes, et al. (Fig EV4A). Further, the top 2,000 genes differentially expressed among soft, stiff, and CSC cells were listed to produce a graphical heat map (Fig 4C), presenting a more tightly clustered genes between the soft and stiff groups relative to those between the CSC and stiff groups, suggesting that softness is better than molecular markers at representing the inherent characteristic of tumorigenic cells. Using a comparable approach, we also

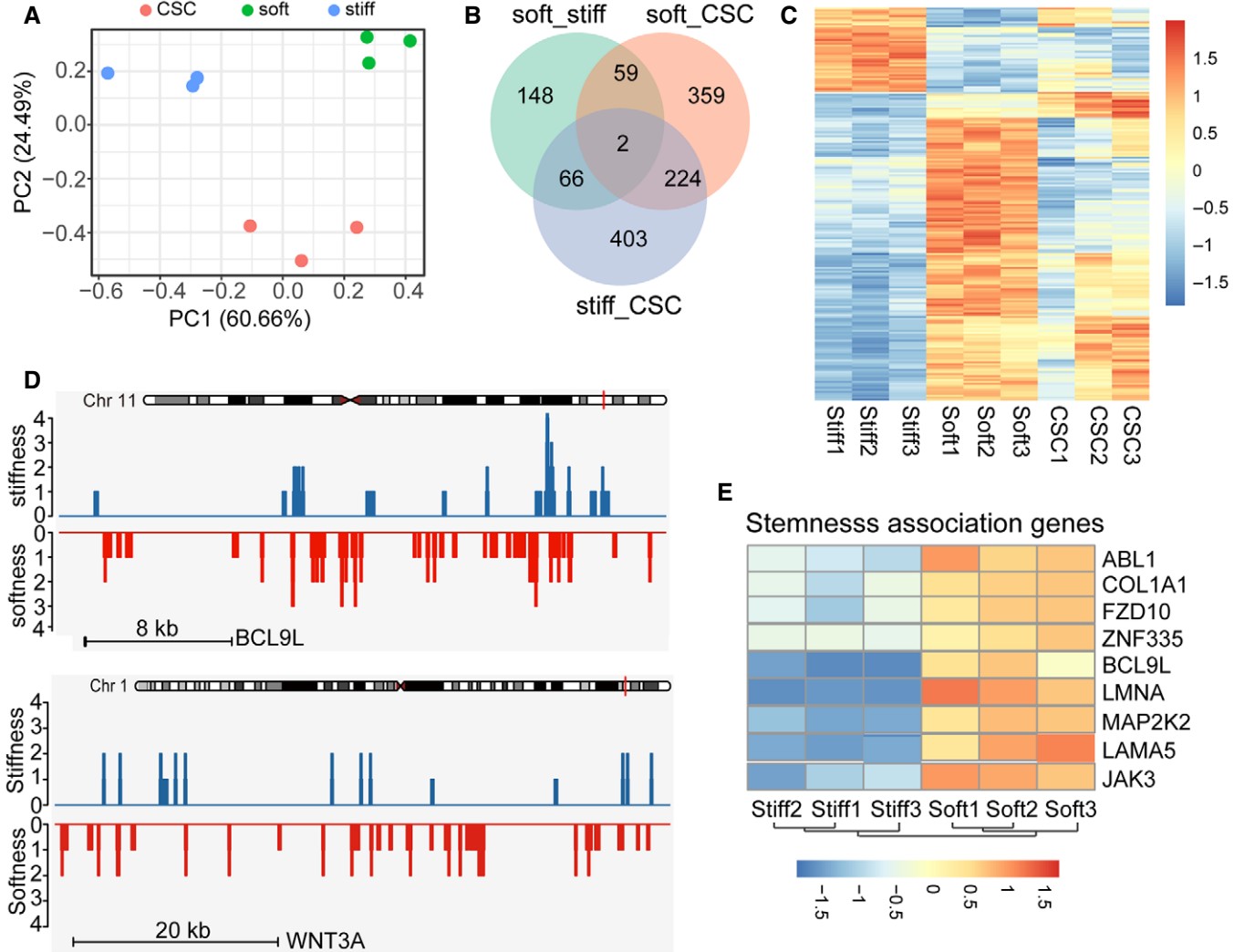

**Figure 4. Stemness associated genes were identified from soft tumor cells.**

A   PCA plot showing the clustering of stiff, soft cells, and cancer stem cells (CSC). The variance is the smallest between stiff and soft on PC2.
B   Venn diagram comparing the hits from stiff, soft, or CSC MCF-7 cells.
C   Heatmap of differentially mRNA-seq expressed genes in stiff, soft, or ALDH[+] (CSC) MCF-7 cells. The unit for the color scale was z-score of log2 expression data shown.
D   ATAC–seq tracks of BCL9L and WNT3A in stiff or soft cells.
E   Heatmap of stemness association genes expression determined by mRNA sequencing comparing stiff and soft MCF-7 cells. $n = 3$. The unit for the color scale was z-score of $\log_2$ expression data shown.

performed a chromatin accessibility analysis through ATAC-seq, which revealed many differentially accessible peaks in soft and stiff tumor cells. Of note, BCL9L, WNT2B, and WNT3A were among the most differentially expressed genes and were also more prominent at the chromatin-opening state in soft tumor cells (Figs 4D and EV4B). In addition, the analysis of stemness-associated genes derived from RNA sequencing also indicated BCL9L as a strong candidate (Fig 4E). Coincidently, Wnt signaling is critical in the regulation of stem cell pluripotency and self-renewal; however, BCL9/BCL9L act additional transcriptional co-activators and form part of the Wnt enhanceosome (van Tienen *et al*, 2017). Thus, we focused on BCL9L (B-cell CLL/lymphoma 9-like), a homologue of BCL9 with functional redundancy.

**BCL9L is a stemness marker for soft tumor cells**

Indeed, we confirmed that the expression of BCL9L was remarkably upregulated in soft breast (4T1 and MCF-7) and melanoma (B16 and MP-1) tumor cells, as evidenced by qPCR, Western blot, and immunostaining (Figs 5A, and EV5A and B). In line with this upregulation of BCL9L, the expression of the total β-catenin and its nuclear form was also enhanced in the soft tumor cells and an enhanced nuclear translocation was observed (Figs 5A, and EV5C and D). Notably, the knockout of BCL9L by CRISPR/Cas9 resulted in the downregulation of β-catenin expression at both the whole cell and nuclear levels in the soft tumor cells (Figs 5B and EV5E). Also, this BCL9L knockout decreased

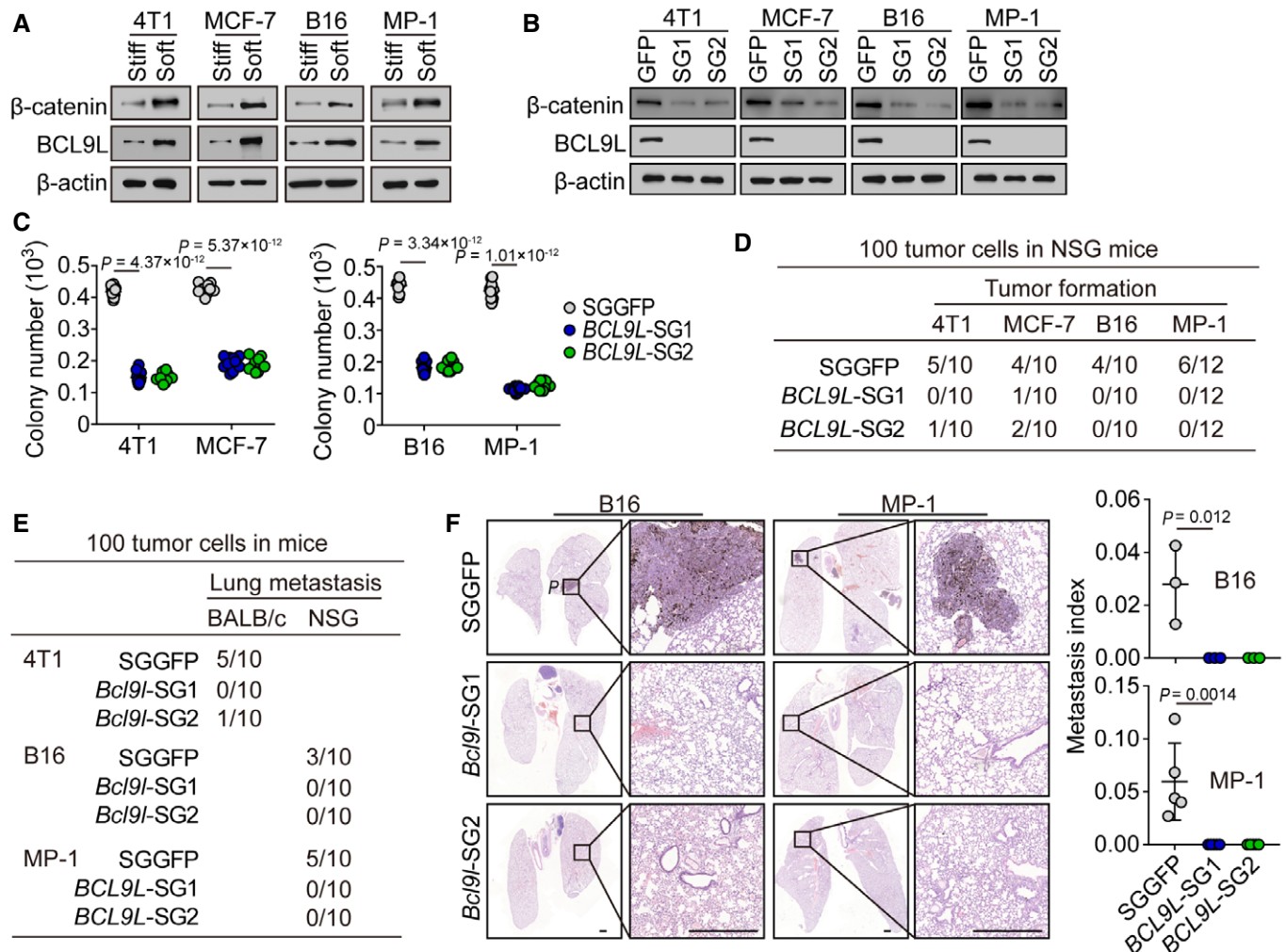

**Figure 5. BCL9L is the key mediator to regulate the stemness and tumorigenicity of soft tumor cells.**

A    The expression of BCL9L or β-catenin in stiff or soft 4T1, MCF-7, B16, or MP-1 cells was determined by Western blot.

B    The expression of β-catenin and BCL9L from SGGFP, *BCL9L*-SGs-4T1, MCF-7, B16, or MP-1 cells were determined by Western blot.

C    500 soft SGGFP or *BCL9L*-SGs- 4T1, MCF-7, B16-F1, or MP-1 cells were seeded in a 90 Pa soft 3D fibrin gel for 6 days. The number of formed colonies was counted. *n* = 10.

D    The soft SGGFP or *BCL9L*-SGs- 4T1, MCF-7, B16, or MP-1 cells (100 cells) were injected into the mammary fat pads (4T1 and MCF-7) or subcutaneous tissue (B16 and MP-1) of NSG mice. Tumor formation was recorded. *n* = 10 or 12 mice as indicated.

E, F  The 100 soft SGGFP or *BCL9L*-SGs- 4T1, B16, or MP-1 cells were injected into the mammary fat pads (4T1) of BALB/c mice or tail vein (B16 and MP-1) of NSG mice. The tumor formation was recorded (E). Six to eight weeks later, the lung sections were H&E stained. The metastatic micronodules in the lung were counted, and the metastasis index was calculated (F). Scale bar, 0.5 mm. *n* = 3 (for B16-F1) or 5 (for MP-1) mice with metastatic tumor.

Data information: Bonferroni test (C and F). The data represent mean ± SD. *n* = 3 independent experiments in (A and B).
Source data are available online for this figure.

colony number and reduced the colony size of tumor cells in the soft 3D fibrin gels (Figs 5C and EV5F). In addition, we found that the expression of stemness genes was upregulated in the soft tumor cells (Nestin, OCT3/4 and SOX2 for 4T1/MCF-7; Nanog, OCT3/4, SOX2, and CD133 for B16F1/MP-1), while the BCL9L knockout reversed this effect (Fig EV5G). To further validate these *in vitro* results *in vivo*, we inoculated mice with 100 soft SGGFP- or *BCL9L*-SGs-tumor cells. Twelve weeks later, we found that the soft SGGFP-tumor cells formed a tumor in the mice; in contrast, no tumors were formed in the BCL9L knockout group (Fig 5D). Also, the BCL9L knockout abrogated the lung

metastasis of soft tumor cells (Figs 5E and F, and EV5H). In line with this *in vivo* result, BCL9L knockout also inhibited soft tumor cell invasion *in vitro* (Fig EV5I). Together, these results suggest that BCL9L is an intrinsic molecular component in soft cells utilized to maintain their stemness.

## Soft tumor cells highly expressed BCL9L in patients

Finally, we sought to validate our findings in clinical patient samples. Using microfluidic devices, we separated soft tumor cells from the breast or colon cancer tissues of patients. We found that the soft

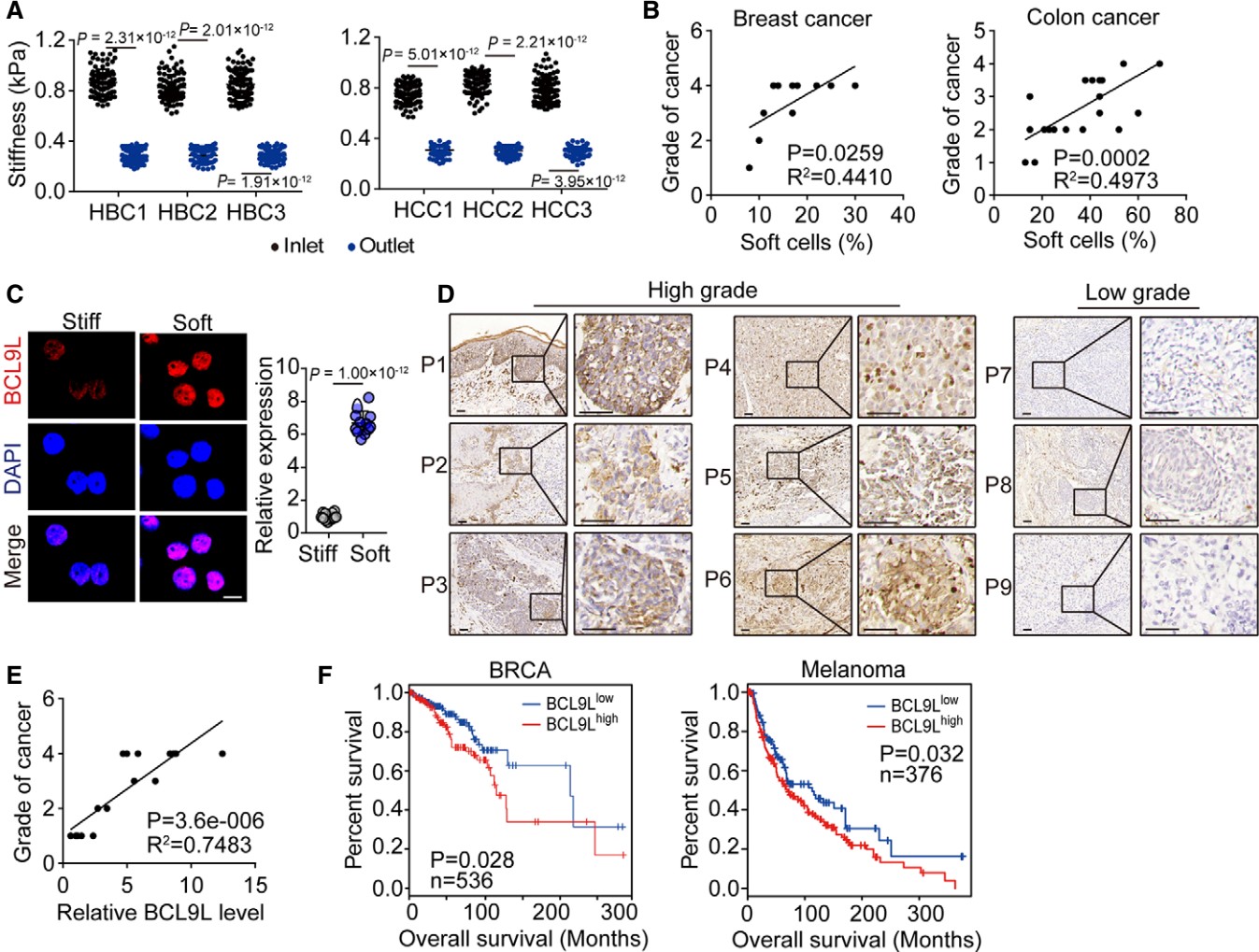

**Figure 6. BCL9L highly expressed in soft tumor cells from cancer patients.**

A The stiffness of primary breast (HBC, $n = 3$) and colon (HCC, $n = 3$) cancer cells screened from the outlet or backflow inlet was measured by AFM. $n = 100$ from each patient.

B The correlation between the percentage of soft primary tumor cells from patients with breast ($n = 11$) or colon ($n = 23$) cancer and the grade of breast or colon cancer, respectively.

C The stiff and soft tumor cells were isolated from 3 patients with breast cancer and stained with anti-BCL9L antibody. The expression of BCL9L was observed under confocal microscope. Scale bar, 10 μm.

D The tissue sections from 6 melanoma patients with high grades of cancer and 3 melanoma patients with low grade of malignancy were immunohistochemical staining with anti-BCL9L antibody. Scale bar, 50 μm.

E The correlation between the expression of BCL9L from 18 patients with melanoma and the grade of melanoma.

F Overall survival compared with the BCL9L level in patients with breast cancer (BRCA, $n = 536$) or melanoma ($n = 376$).

Data information: Mann–Whitney test (A), paired Student's *t*-test (C), Pearson's correlation test (B and E) or Log-rank survival analysis (F). The data represent mean ± SD.

primary breast or colon cancer cells displayed a cellular stiffness of around 0.3 kPa, while their separated stiff counterparts were over 0.7 kPa (Fig 6A). Moreover, the percentage of soft primary tumor cells isolated from patients with colon ($n = 23$) or breast ($n = 11$) cancer was strongly correlated with the tumor pathological grade (Fig 6B). Then, we detected the expression of BCL9L in soft and stiff tumor cells isolated from patients with breast cancer. We found that BCL9L levels were much higher in soft tumor cells compared with stiff ones (Fig 6C). Further screening of BCL9L in 16 patients

with melanoma showed that BCL9L was highly expressed in melanoma tissues and tumor BCL9L levels was positively correlated with the stage of melanoma (Fig 6D and E). Consistently, the high expression of BCL9L in tumor tissues was correlated with a poorer prognosis of patients with melanoma, breast cancer, and other cancer types such as pancreatic cancer, lung cancer, and liver cancer (Figs 6F and EV5J). These data suggested that mechanical softness might be a potential prognostic indicator for patients with cancer.

# Discussion

The CSC theory remains controversial regarding their origin, frequency, phenotype, and function (Clevers, 2011; Kaiser, 2015; Wang *et al*, 2015). Although current CSC markers enable the development of clinical diagnostics and CSC-based therapies (Keysar & Jimeno, 2010; Medema, 2013), such markers can also be detected, more or less, on normal stem cells, differentiated cancer cells, or even normal tissues (Ginestier *et al*, 2007; Gregorieff *et al*, 2015). Furthermore, CSC markers are very unstable. For instance, both CD133$^+$ and CD133$^-$ tumor cells have been identified as CSCs (Singh *et al*, 2004; Beier *et al*, 2007; Shmelkov *et al*, 2008). Also, both CD34$^+$CD38$^-$ and CD34$^+$CD38$^+$ AML cells have displayed tumorigenic activity (Taussig *et al*, 2008; Taussig *et al*, 2010). Unlike the variability of biochemical molecules, the physical trait of a cell is more stable. In this study, we provide evidence that cellular softness can act as a universal marker to define CSCs.

Softness confers cells with the ability to deform. Unlike previously reported microfluidic chips, used to isolate soft cells (Mohamed *et al*, 2009; Guo *et al*, 2012), we have developed a chip with single-stream features, such as a longer channel in a limited area, efficient collection of cells by forward and reverse flows, and convenient single-cell observation by microscopy. The use of a silicon mold for device fabrication ensures process accuracy and the convenience for producing multiple duplicates of polydimethylsiloxane microfluidic chips, which is important for good experimental control. Using this device, we isolated both soft and stiff tumor cells. Notably, only the isolated soft tumor cells were able to form colonies in soft 3D fibrin gels but the stiff ones could not form colonies. Moreover, conventional marker-based CSCs, such as CD133$^+$, ALDH1$^+$, or SP$^+$ CSCs, isolated from melanoma or breast cancer cells, contain both soft and stiff subsets; and only the soft subpopulation has the ability to form a tumor. In addition to tumor formation, an important feature for CSCs lies in the ability to metastasize to distant organ(s). Consistently, the soft tumor cells display the capacity of invasion and metastasis; however, the stiff ones demonstrate poor capacity for metastasis.

Published reports have shown a positive association between increased tissue stiffness and aggressive cancer behavior, prompting to propose a model of cancer progression that depends on static or dynamic tumor tissue stiffening (Paszek *et al*, 2005; Wei *et al*, 2015; Ondeck *et al*, 2019). However, there has been no evidence to support the notion that the cells coming out of the stiffened tumor stroma are tumorigenic and metastatic cancer cells. It has also been unclear whether the tumorigenic cells are stiff cells or soft cells. Despite the overall stiffness, local microenvironments for tumor stiffness are highly heterogeneous (Plodinec *et al*, 2012). Increased tissue stiffness may be attributed to more extracellular matrices, which are likely to limit blood vessel distribution and lead to tumor hypoxia, a common phenomenon in tumor microenvironments. It is known that hypoxia in primary tumors is associated with an increased metastasis and a worse prognosis in patients with cancer (Erler *et al*, 2006; Gilkes *et al*, 2014; Rankin & Giaccia, 2016). Recently, we demonstrated that hypoxia promote human breast tumor-repopulating cell development (Tang *et al*, 2019). Of note, hypoxic areas may be very soft due to local tissue necrosis and matrix degradation. Thus, increased tissue stiffness may result in more soft tumor cells at the hypoxic sites, favoring an aggressive

cancer behavior. Consistently, studies by Superfine *et al* showed that cancer cells with the highest migratory and invasive potential are five times less stiff than cells with the lowest migration and invasion potential (Swaminathan *et al*, 2011). Our results in this study strongly suggest that only the intrinsically soft tumor cells are able to be highly tumorigenic and metastatic in animal models. Our findings are consistent with the reports that metastatic tumor cells are much softer that non-metastatic cancer cells (Guck *et al*, 2005; Cross *et al*, 2007; Plodinec *et al*, 2012; Xu *et al*, 2012). Our soft tumor cell data are also consistent with the finding that weakly adherent tumor cells are more migratory and metastatic (Hope *et al*, 2004). In addition to hypoxia, it is possible that local tumor tissue stiffening may promote the epithelial–mesenchymal transition and thus the stiffening of tumors cells might facilitate a subpopulation of tumor cells to differentiate and to migrate out of the tumor stroma. However, it is only those soft undifferentiated tumor cells that also come out of the stroma which are able to metastasize and proliferate at the secondary sites, as we have proposed previously (Tan *et al*, 2014). Nevertheless, whether the soft tumor cells or the stiff tumor cells convey metastatic and tumorigenic potential in human patients needs to be examined carefully in the future.

CSC theory implicates the use of CSC markers to predict patient prognosis. However, existing evidence is conflicting. Some studies show that CSC markers are correlated with a poor prognosis and overall survival (Ginestier *et al*, 2007; Iinuma *et al*, 2011; Stavropoulou *et al*, 2016), but some studies indicate that CSC markers like CD133 and ALDH1 have no correlation with prognosis (Lugli *et al*, 2010; Wakamatsu *et al*, 2012; Kapucuoğlu *et al*, 2015; Miller *et al*, 2017). This might be due to the notion that marker-based CSCs contain both soft and stiff tumor cells. Thus, the use of cellular softness may be a potential physical marker used to predict cancer patient prognosis. An important finding in this study is that BCL9L is identified as a biological marker for soft tumor cells to distinguish them from their stiff counterparts. Wnt/β-catenin signaling plays a crucial role in the regulation of the pluripotency and self-renewal of stem cells. To exert its function, β-catenin enters the nucleus, where β-catenin triggers Wnt-mediated transcription in association with transcription factors TCF/Lef. However, BCL9L acts as an additional transcriptional co-activator and form part of the Wnt enhanceosome. BCL9L is upregulated in soft tumor cells but very weakly expressed in stiff tumor cells. More importantly, in our small cohort study, we indeed observed that BCL9L expression is related to the prognosis of patients with cancer. Based on these analyses, we propose that BCL9L might function as a useful marker to predict cancer patient prognosis.

# Materials and Methods

### Animals and cell lines

Six-week-old female C57BL/6, BALB/c, and NOD/SCID IL-2Rγ-null mice (NSG) were purchased from the Center of Medical Experimental Animals of the Chinese Academy of Medical Science (Beijing, China). These animals were maintained in the Animal Facilities of the Chinese Academy of Medical Science under pathogen-free conditions. All studies involving mice were approved by the Animal Care and Use Committee of the Chinese Academy of Medical Science.

Murine B16 melanoma and 4T1 breast cancer, and human MCF-7 breast cancer cell lines were purchased from the China Center for Type Culture Collection (Beijing, China). Primary tumor cells isolated from human melanoma (MP-1) and human colorectal cancer (HCC-5 and HCC-6) tissue were grown in RPMI1640 medium (Gibco, USA). B16 and MCF-7 cells were cultured in DMEM (Gibco, USA), and 4T1 was grown in RPMI1640 medium. All media was supplemented with 10% fetal bovine serum (FBS) (Gibco, USA) and 2 mM L-glutamine (Gibco, USA). All cells were grown at 37°C in a 5% $CO_2$ incubator.

## Human samples

Colon or breast cancer tissues were obtained from patients at the National Cancer Center/Cancer Hospital. Ethical permission was granted by the Clinical Trial Ethics Committee of National Cancer Center/Cancer Hospital. The paraffin embedding tumor tissues of patients with melanoma were obtained from the Department of Pathology, Cancer Hospital of Yunnan Province. Ethical review was granted by the Institutional Ethics Committee of Cancer Hospital of Yunnan Province. The clinical features of the patients are listed in Tables EV2–EV4.

## Reagents, materials, and equipment

Cytochalasin D (CD, Cat. PHZ1063) and Jasplakinolide (Jas, Cat. J4580) were purchased from Invivogen and Sigma, respectively. Silicon wafers (single crystal, n-type doping) were purchased from No.46 Research Institute of China Electronics Technology Corporation (CETC46, Tianjin, China). RZJ-304 photoresist used in the lithography was from Ruihong Comp. (Shanghai, China). Sylgard 184 silicone elastomer (polydimethylsiloxane, PDMS), including base and curing agent, was purchased from Dow Corning Corp. (Midland, USA). The corona plasma treater (BD-20AC) for PDMS/glass bonding was from ETP (Electro-Technic Products, Chicago, USA). The syringe pump (Fusion 200) used in this work was purchased from Chemyx Inc. (Stafford, USA). The plastic tube was custom-made by Nantong Yinuo Precision Plastic Pipe Co. LTD. (Nantong, Jiangsu, China). The syringes were purchased from Shanghai Gaoge Industrial and Trading Co. LTD. (Shanghai, China).

## Fabrication of microfluidic chips

Fabrication of the PDMS soft-lithography mold started with a 4-inch silicon wafer. First, 3,000 Å $SiO_2$ was formed by a thermal oxidation. Then, the first lithography step was conducted to define the flow channel and microweir structure followed by selective removal of the exposed $SiO_2$ by BHF (buffered hydrofluoric acid) bath. Next, the second lithography and RIE (reacting ion etching) of $SiO_2$ were successively conducted, as an ingenious design for elimination of alignment errors from two lithography steps. Afterwards, a first DRIE (deep reactive ion etching) was conducted with the patterned photoresist (from the second lithography) as the etching mask to form the initial depth (flow channel height minuses microweir height). After stripping the photoresist, the second DRIE (etching depth equaling to the microweir height) was applied directly to form the two-depth silicon structure with the pre-defined $SiO_2$ as the etching mask. The final silicon mold was obtained after removing $SiO_2$

in a BHF bath. Next, PDMS pre-polymer was prepared by mixing the base and the curing agent with a weight ratio of 10:1. The mixture was degassed for 20 min in a vacuum desiccator. Then, the pre-polymer was poured onto the surface pre-treated (release agent coating) silicon mold and cured in an oven at 70°C for 60 min. After released from the mold, the PDMS were punched through to function as inlet/out and then bonded with a glass slide by an $O_2$ plasma treatment of 10 s. In this study, chips with four different designs, including different lengths, widths of microweir structure, spaces between microweir structures, and total numbers of microweir structures were designed and investigated.

## Cell screening through the microfluidic chips

The microfluidic chip was connected to a syringe pump through plastic tubes for sample loading. The chip was first rinsed with 75% ethanol and 1% ($w/v$) bovine serum albumin (BSA) sequentially. Then, the cell suspension (density ranging from $1 \times 10^4$ to $2 \times 10^4$/ml) was injected into the chip through the inlet, driven by a syringe pump with a flow rate at 10 μl/min. After the cell solution loading, the cells getting to the outlet were collected as soft cells. For the screening of stiff cells, the residual cells in the Chip (cells that did not pass through the microweir structures) were driven back into the inlet by a reverse flow and collected as stiff cells.

## Animal experiments protocol

All studies involving mice were approved by the Animal Care and Use Committee of the CAMS. All the animals were allocated randomly. For tumor formation, the isolated stiff or soft 4T1 or MCF-7 cells were injected into the mammary fat pads of BALB/c or NSG mice, and B16 or MP-1 cells were subcutaneously injected into the right flank of C57BL/6 or NSG mice. Mice were examined weekly for tumors by observation and palpation. Eight to twelve weeks later, the percentage of tumor formation was calculated. For the generation of lung metastatic models, C57BL/6 or NSG mice were injected with isolated soft or stiff B16 or MP-1 cells by tail vein injection. Six weeks later, H&E and immunohistochemistry staining of lungs from mice were applied to evaluate metastatic ability.

## Analysis of cellular stiffness by atomic force microscopy

Cell stiffness was measured by using a BioScope Resolve (Bruker, Santa Barbara, USA) AFM. The cantilevers for AFM were with a nominal spring constant of 0.5 Nm-1. The AFM imaging was recorded at room temperature and the scan rate was 1.00 Hz, with tip velocity of 100 μm/s. The Peak Force Setpoint for Feedback was set to 0.6 V. The peak force frequency and peak force amplitude were set to 1 kHz and 300 nm, respectively. Hundreds of cells were imaged and the cellular stiffness was measured and analyzed with Nanoscope Analysis 1.9 Software (Bruker, Karlsruhe, Germany).

## Real-time PCR

TRIzol (Invitrogen, Cat. 15596018) was used to extract total RNA from cells which was then transcribed to cDNA using a high capacity cDNA reverse transcription kit (Applied Biosystems, Cat. 4368813). Real-time PCR was performed using ABI stepone plus

(Applied Biosystems, CA, USA). The primer sequences are shown as follows: *Gapdh*, AGGTCGGTGTGAACGGATTTG (sense) and TGTA-GACCATGTAGTTG AGGTCA (antisense); *GAPDH*, ACAACTTTGG-TATCGTGGAAGG (sense) and GCCATCACGCCACAGTTTC (antisense); *Bcl9l*, CGCGAGAGGAGTGTGTCTG (sense) and CCAT TCGTCCCCACTGTACG (antisense); *BCL9L*, TCTCGCCTAGC AACT-CAAGTC (sense) and GAGCACCATTCGTCCCCAC (antisense); *Ctnnb1*, ATGGAGCCGGACAGAAAAGC (sense) and CTTGCCACT-CAGGGAAGGA (antisense); *CTNNB1*, CATCTACACAGTTTGATGC TGCT (sense) and GCAGTTTT GTCAGTTCAGGGA (antisense). Values are means ± SD from three independent experiments which were performed in duplicate. Statistical comparisons among groups were performed using one-way ANOVA or Student's *t*-test. Values of all parameters were considered statistically significant difference at a value of $P < 0.05$.

## Generation of CRISPR-Cas9 knockout cell lines

For construction of the stable knockdown of BCL9L-B16, 4T1, or MCF-7 cells, the following SGRNAs targeting BCL9L were used: SGGFP, CACCGGGGCGAGGAG CTGTTCACCG (sense) and AAA CCGGTGAACAGCTC CTCGCCCC (antisense); *BCL9L*-SGRNA1, TGACCAATCATGGCAAGACA (sense) and TGTCTTGCCAT GATT GGTCA (antisense); *BCL9L*-SGRNA2, CCAAGGACCCACCTGCAACG (sense) and CGTTGCAGGTGGGTCCTTGG (antisense); *Bcl9l*-SGRNA1, CCAGG TTACCCCACCCCAGG (sense) and CCTGGGGTG GGGTAACC TGG (antisense); *Bcl9l*-SGRNA2, AGTCCACCGCTG TCCCCTCG (sense) and CGAGGGGACAGC GGTGGACT (antisense). These SGRNAs were cloned into the pL-CRISPR.EFS.RFP vector plasmid (addgene, #57819) and transfected HEK 293T cells together with the packing plasmids psPAX2 and pMD2.G. Forty-eight hours later, the lentivirus was harvested and concentrated to infect B16, 4T1, or MCF-7 cells together with polybrene at a final concentration of 8 μg/ml. Two days later, RFP-positive cells were sorted by flow cytometry using the BD Biosciences FACSAria III. The candidate knockout cells were verified by Western blot.

## Western blotting

Cells were collected, lysed in M2 lysis buffer, and sonicated. The protein concentrations were determined by the BCA kit (Applygen Technologies Inc., China). Then, the protein was run on an SDS–PAGE gel and transferred to nitrocellulose membrane. Nitrocellulose membranes were blocked in 5% bovine serum albumin (BSA) and probed with antibodies overnight: anti-β-actin (Cell signaling technology, Cat.: 3700S; clone: 8H10D10); BCL9L (Thermo fisher, Cat.: PA5-21111), and anti-β-catenin (Cell signaling technology, Cat.: 8480S; clone: D10A8). Secondary antibodies conjugated to horse-radish peroxidase were followed by enhanced chemiluminescence (Thermo fisher, MA). Results were confirmed by at least three independent experiments.

## Hoechst 33342 staining and ALDH⁺, CD133⁺, or SP tumor cells sorting

The tumor cells were resuspended ($10^6$ cells/ml) in Hank's balanced salt solution (HBSS) and incubated for 30 min at 37°C with 1 μg/ml Hoechst 33342 (Thermo fisher, Cat.: H1399). Then, these suspended cells were stained with PE-conjugated anti-CD133 antibody (1:100; Biolegend, Cat.: 393904) at room temperature for 30 min. After being washed with ice-cold HBSS, these cells were filtered through a 40 μm cell strainer (JET Biofil, China) to obtain a single-cell suspension for sorting on a BD FACSARIA III (BD Bioscience, NJ, USA). The ALDH⁺ cells were sorted by using ALDEFLUOR Kit (Stemcell Technologies, Cat.: 01700) according to the supplier's instruction.

## Immunofluorescence

Cells were fixed in 4% polyoxymethylene and permeabilized with 0.5% Triton X-100 at 4°C for 10 min. Then, these cells were blocked with 5% BSA for 20 min at room temperature. After incubation with anti-β-catenin (1:1,000, Cell signaling technology, Cat.: 8480S) or BCL9L (1:1,000, Thermo fisher, Cat.: PA5-21111) at 4°C overnight, cells were washed and incubated sequentially with HRP-conjugated secondary antibodies for 1 h at room temperature. At last, the slides were counterstained with DAPI and mounted for confocal analysis. The intensity of immunofluorescence was analyzed by ImageJ 9.0 software.

## Wound-healing assay

The monolayer cells in 6-well plate were scraped in a straight line with a 10-μl pipette tip to produce a wound. After being washed with ice-cold HBSS, cells were incubated in medium containing 1% fetal bovine serum (FBS). Photographs of the scratch were taken at 0 and 24 h after wounding using the Olympus-inverted microscope. Gap width at 0 h was set to 1. Multiple defined sites along the scratch were measured. Data are shown as the average of three independent experiments.

## Migration and invasion assay

For cell migration assay, $2–10 × 10^4$ cells in 200 μl serum-free medium were plated in an 8.0-μm, 24-well Hanging Insert (Millipore, Cat.: MCEP24H48), and 400 μl medium containing 10% FBS was added to the lower chamber of a transwell dish. After incubation for 24–48 h, non-migrating cells were removed from the upper surface of the membrane, and cells that migrated through the 8 μm pore membrane were fixed with 4% paraformaldehyde and stained with 0.5% crystal violet. The migrating cells were photographed and counted using an inverted microscope. For the cell invasion assay, ECMatrix™-coated chambers (Millipore, Cat.: ECM550) were used according to the supplier's instruction. Briefly, the cell suspension containing $0.5–1.0 × 10^6$ cells/ml in serum-free media was added to each insert, and 500 μl medium containing 10% FBS was added to the lower chamber of a transwell dish. After incubation for 24–72 h, the non-invading cells as well as the ECMatrix gel were removed gently from the interior of the inserts by using a cotton-tipped swab. The invading cells were fixed with 4% paraformaldehyde and stained with 0.5% crystal violet for 20 min, and photographed and counted using an inverted microscope.

## Isolation of primary tumor cells

The surgically removed tumor tissues were cut into small pieces of 1–3 mm, minced, and incubated for 1 h at 37°C under

continuous rotation with RPMI 1640 medium supplemented with collagenase type Ⅳ (Sigma, 32 μg/ml, Cat. C5138), hyaluronidase (Sigma, 500 μg/ml, Cat. H1136), and DNAase I (Sigma, 5 μg/ml, Cat. 11284932001). After the digestion, single-cell suspensions were obtained by filtering through a 70-μm cell strainer before being pelleted by centrifugation and then erythrocytes (RBC) were removed by exposed to RBC lysis buffer. Then, single-cell suspensions were magnetically labeled using the Non-Tumor Cell Depletion Cocktails microbeads contained in the Tumor Cell Isolation Kit (Miltenyi Biotec, Cat. 130-108-339), according to the supplier's instruction. Finally, the unlabeled tumor cells flow-through the LC column in the magnetic field of MACS® Separator were collected for subsequent analysis. Tumor cells were isolated to a purity of > 95%, regardless of the starting frequency of tumor cells.

### Quantification and statistical analysis

All experiments were performed at least three times. Results are expressed as mean ± SD as indicated and analyzed by Student's *t*-test followed by two-tailed paired *t*-test or Mann–Whitney test or one-way ANOVA followed by Bonferroni or Kruskal–Wallis test as indicated. The $P$ value < 0.05 was considered statistically significant. The analysis was conducted using the GraphPad 6.0 software. The survival rates were evaluated by the Log-rank test.

## Data availability

The datasets produced in this study are available in the following databases: RNA-Seq and ATAC-seq data: The National Genomics Data Center (NGDC) Genome Sequence Archive PRJCA003394 (https://bigd.big.ac.cn/gsa/s/3L1LdBT6).

**Expanded View** for this article is available online.

### Acknowledgements
This work was supported by National Natural Science Foundation of China (81788101, 81530080, 81773062, 91942314), Chinese Academy of Medical Sciences (CAMS) Initiative for Innovative Medicine (CAMS-I2M) 2017-I2M-1-001, and 2016-I2M-1-007.

### Author contributions
Project conception: BH and YYL; Experiments: JLv, YPL, FC, JLi, YZ, TZ, NZ, CL, ZW, LM, QZ, XL, ML, KT, JM, HZ, YYL, and JX; Methodology development: YPL, JLv, YYL, YF, HZ, and NW; Data analysis: BH, YPL, JLv, YYL, and JLi; Manuscript writing: BH, YYL, and JLv.

### Conflict of interest
The authors declare that they have no conflict of interest.

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
