## [Review Process File · The EMBO Journal]

Cell softness regulates tumorigenicity and stemness of cancer cells

Bo Huang, Jiadi Lv, Yaoping Liu, Feiran Cheng, Jiping Li, Yabo Zhou, Tianzhen Zhang, Nannan Zhou, Cong Li, Zhenfeng Wang, Longfei Ma, Mengyu Liu, Qiang Zhu, Xiaohan Liu, Ke Tang, Jingwei Ma, Huafeng Zhang, Jing Xie, Yi Fang, Haizeng Zhang, Ning Wang, and Yuying Liu

DOI: [10.15252/emboj.2020106123](https://doi.org/10.15252/emboj.2020106123)

Corresponding authors: *Bo Huang* (tjhuangbo@hotmail.com) , *Yuying Liu* (13161773902@163.com)

Review Timeline:

Submission Date:	2nd Jul 20
Editorial Decision:	13th Aug 20
Revision Received:	9th Sep 20
Editorial Decision:	1st Oct 20
Revision Received:	7th Oct 20
Accepted:	23rd Oct 20

Editor: Ieva Gailite

Transaction Report:

Thank you for submitting your manuscript for consideration by The EMBO Journal. We have now received two referee reports on your manuscript, which are included below for your information. Since we have not been able to obtain a third report in a reasonable time frame, I am taking the decision based on the comments at hand to avoid further delays.

As you will see from the comments, both reviewers appreciate the topic and the quality of the study. However, they also indicate a number of concerns that would have to be addressed and clarified before they can support publication of the manuscript. From my side, I find the reviewer comments generally reasonable. Therefore, I would invite you to address the concerns raised by both reviewers in a revised manuscript.

I should add that it is The EMBO Journal policy to allow only a single major round of revision and that it is therefore important to resolve the main concerns at this stage. We are aware that many laboratories cannot function at full efficiency during the current COVID-19/SARS-CoV-2 pandemic, and I would be happy to discuss the revision in more detail via email or phone/videoconferencing.

We have extended our 'scooping protection policy' beyond the usual 3 month revision timeline to cover the period required for a full revision to address the essential experimental issues. This means that competing manuscripts published during revision period will not negatively impact on our assessment of the conceptual advance presented by your study. Please contact me if you see a paper with related content published elsewhere to discuss the appropriate course of action.

When preparing your letter of response to the referees' comments, please bear in mind that this will form part of the Review Process File, and will therefore be available online to the community. For more details on our Transparent Editorial Process, please visit our website:

<https://www.embopress.org/page/journal/14602075/authorguide#transparentprocess>. Please also see the attached instructions for further guidelines on preparation of the revised manuscript.

Please feel free to contact me if you have any further questions regarding the revision. Thank you for the opportunity to consider your work for publication. I look forward to receiving the revised manuscript.

Referee #1:

This study presents a microfluidic device for separating cells according to mechanical compliance ("softness") and goes on to show that softer cells are more tumorigenic, with both softness and tumorigenicity being regulated by BCL-9. This is a novel and well-designed study that nicely integrates microdevices, mouse models, analysis of patient samples, and mechanistic culture studies. It goes beyond many past studies in its rigorous molecular characterization of soft vs stiff cells and use of multiple experimental paradigms. My main criticism is that it's not clear that softness is directly responsible for driving differences in malignancy, versus being a more passive biomarker. But either possibility would be significant, and the work is likely to be of great interest in any case. I just have a few minor comments for the authors to address, after which I would recommend publication. I don't view any of my suggested experiments as truly essential to publication.

1. It would be helpful to have a bit more background on how the various microweir design parameters would theoretically be expected to affect stiffness-sorting. Were these dimensions arrived at in an ad hoc fashion, or did the authors do computational design? If so, how well do the experiments match the predictions?
2. In Fig EV1 there seems to be disconnect between the text and fig for the Jas result. The text indicates that Jas stiffens cells and produces fewer cells at the outlet. However, the figure is missing any indication that Jas-induced differences are statistically significant. Are symbols missing from the figure, or is the text incorrect?
3. The choice of ANOVA and t-tests for many of the statistical comparisons is puzzling given that the distributions seem so non-Gaussian (e.g. EV1A, EV3E). This is more than an academic detail given the enormous spread in some of these data sets. Could the authors justify this decision, and do the statistical differences hold when tests for non-normal distributions are used (e.g. Mann-Whitney or Kruskal-Wallis)?
4. Do the soft cells remain uniformly soft after culture in the fibrin gels, or are softness distributions re-established? The former would suggest that the softness cues are functionally important for colony formation (and later invasion), whereas the latter would suggest that softness is more of a selection marker that happens to capture the most tumorigenic cells.
5. There appears to be some conceptual overlap from earlier studies involving some of these authors, e.g. Tan et al Nat Comm 5: 4619 (2014). The authors should incorporate this paper more explicitly in their introduction and explain more clearly the new advances here.
6. Based on publicly available tumor sequence databases (e.g. TCGA), is BCL9L expression associated with malignancy and/or poor prognosis?

Referee #2:

In this study, the investigators explore the potential of cellular softness as a marker of tumor-initiating and metastatic cancer cells. They use a specific microfluidic chip to isolate stiff and soft cancer cells from different malignancies, and this is followed by characterization of the isolated cells in vitro and in vivo. The results indicate that soft cancer cells have tumor initiating capacity and are more metastatic than stiff cells. Interestingly, this is independent of previously identified cancer stem cell (CSC) markers. The authors use transcriptomic analysis to identify BCL9L, a co-factor of beta-catenin and regulator of Wnt signaling, as a functional mediator of aggressivity in soft cancer cells. Finally, the investigators analyze BCL9L expression in clinical cancer samples and identify association between BCL9L and poor patient outcome. This is a very interesting and overall thorough study. However, I do have some critical points that need to be addressed. Details below.

1. The investigators characterize soft cancer cells and demonstrate that this property is an excellent marker of tumor-initiating ability and cancer aggressiveness across different cancer types. Growing evidence indicates that tumor-initiating properties can be surprisingly plastic and context dependent, suggesting that cancer cells may gain or lose these properties depending on the microenvironment. The authors state that physical traits may be more stable than expression of CSC marker genes. With this in mind, I think it would be important to show that different levels of stiffness are indeed stable properties. What happens when selected soft cells are grown for some time in culture? Do they maintain the soft phenotype? What about growth of selected cells in stiff matrix rather than soft matrix? Does it influence the cellular stiffness.

2. Following up on the previous point, is it fair to use soft matrix when colony formation is measured to compare soft and stiff cells? How would the comparison turn out in stiff matrix? Would this now benefit stiff cells?

3. In Figure 6B and 6E, the authors address correlation between tumor grade and the percentage of soft cells or the levels of BCL9L in patient samples. However, the graphs presented are not appropriate to address this. A simple correlation plot with the parameters on x and y axis would be more appropriate. For statistical analysis, the investigators use Pearson's correlation test and thus the correlation coefficient R should be included.

4. It is interesting to see how much better of a marker the cellular softness can be compared to previously recognized CSC markers (Figure 3 and EV3). Notably, a number of the CSC markers (ALDH, Side Population and CD133) were significantly associated with increased softness. Moreover, the combination of ALDH and softness, as markers to isolate cancer cells, modestly improved tumor initiation in mice compared to softness alone (Fig 3B). Not all soft cells express CSC markers and not all CSC marker positive cells are soft. Therefore, it would be insightful to explore the combination of softness and CSC markers a bit further. Does the combination of softness and other CSC markers select for more enhanced tumor-initiation or metastatic ability compared to softness alone. Further analysis of the double positive cells (soft + CSC markers) is warranted.

5. The results presented in this manuscript should be discussed further in the context of the large amount of studies showing positive association between increased tissue stiffness and aggressive cancer behavior. Here the authors show that soft cells are more mobile and invasive. However, substantial evidence suggests that cross-linked collagen mediated by LOX family members and leading to increased stiffness is associated with increased invasion and aggressive cancer behavior.
6. It is not clear to this reviewer what is being compared in Figure 6D.
7. For BCL9L knockout, CRISPR.EFS.GFP positive cells were selected by flow cytometry based on GFP expression. However, if guide RNA against GFP was used as a control, would this not affect selection of CRISPR.EFS.GFP positive cells?
8. It is not clear how the metastasis index is derived. What does it mean specifically?
9. There is a discordance between text and Figure 2E legend. The figure legend states "(E) the same as (B)", essentially primary tumor analysis. However, the main text states that metastatic colonization of the lungs was analyzed.
10. Total beta-catenin expression is not a good marker of active Wnt signaling. Nuclear beta-catenin or a reporter are more reliable.

RESPONSES TO REVIEWERS

We would like to express our sincere thanks to the reviewers for their critical and constructive comments. We respond point-by-point to each of their comments and criticisms. We feel that their comments have helped us on substantially improving and strengthening the manuscript and clarifying some issues.

RESPONSE TO REVIEWER #1:

This study presents a microfluidic device for separating cells according to mechanical compliance ("softness") and goes on to show that softer cells are more tumorigenic, with both softness and tumorigenicity being regulated by BCL-9. This is a novel and well-designed study that nicely integrates microdevices, mouse models, analysis of patient samples, and mechanistic culture studies. It goes beyond many past studies in its rigorous molecular characterization of soft vs stiff cells and use of multiple experimental paradigms. My main criticism is that it's not clear that softness is directly responsible for driving differences in malignancy, versus being a more passive biomarker. But either possibility would be significant, and the work is likely to be of great interest in any case. I just have a few minor comments for the authors to address, after which I would recommend publication. I don't view any of my suggested experiments as truly essential to publication.

1. It would be helpful to have a bit more background on how the various microweir design parameters would theoretically be expected to affect stiffness-sorting. Were these dimensions arrived at in an ad hoc fashion, or did the authors do computational design? If so, how well do the experiments match the predictions?

Response:

We thank the reviewer's pertinent comment. Tumor cells usually have the size around 20-25 μm (*Anal Chem.* 2010;82:6629-35; *Int J Cancer.* 2016;138:2894-904). In our preliminary experiments, we actually tested different size of gaps, including 10 μm , 15 μm and 20 μm . When bulk tumor cells passed through the channel, we found that the 10 μm -microfluidic chips yielded very few soft cells, while 20 μm gap allowed many stiff cells to pass through the channel (unpublished data). Therefore, we designed a 15 μm gap to isolate soft tumor cells. On the other hand, we designed the 18 μm gap to allow soft passing, and isolate stiff tumor cells through inversely washing the channels.

In addition, we selected the 40 μm height of microweir based on previous studies (*Proc Natl Acad Sci.*2012;109: 18707-18712; *J Chromatogr A.* 2009;1216: 8289-8295), thus allowing tumor cells to pass through the channel in a single cell form. We also referred to the

previous studies and tested different widths of microweir structure, spaces between microweir structures, and total numbers of microweir structures.

According to the reviewer's suggestion, we added the tumor cell size background and the references in the revised manuscript, page 5 line 15.

2. In Fig EV1 there seems to be disconnect between the text and fig for the Jas result. The text indicates that Jas stiffens cells and produces fewer cells at the outlet. However, the figure is missing any indication that Jas-induced differences are statistically significant. Are symbols missing from the figure, or is the text incorrect?

Response:

We thank the reviewer's indicating this statistical issue for Fig. EV1. In the revised manuscript, we added the P value between the PBS control and the Jas group.

3. The choice of ANOVA and t-tests for many of the statistical comparisons is puzzling given that the distributions seem so non-Gaussian (e.g. EV1A, EV3E). This is more than an academic detail given the enormous spread in some of these data sets. Could the authors justify this decision, and do the statistical differences hold when tests for non-normal distributions are used (e.g. Mann-Whitney or Kruskal-Wallis)?

Response:

We thank the reviewer for indicating the statistical issue. Accordingly, we did the statistical analysis by using the Mann-Whitney or Kruskal-Wallis method and similar results were obtained.

According the reviewer's suggestion, we revised the corresponding figure legends and material methods, in the revised manuscript, page 34 line 22.

4. Do the soft cells remain uniformly soft after culture in the fibrin gels, or are softness distributions re-established? The former would suggest that the softness cues are functionally important for colony formation (and later invasion), whereas the latter would suggest that softness is more of a selection marker that happens to capture the most tumorigenic cells.

Response:

We thank the reviewer's comments. Stiffness is an inherent feature of a cell, which is mainly provided by F-actin filaments (*Science* 1993;260:1124-7). Different cell types display varying levels of stiffness, which matches the stiffness of the local extracellular matrix, allowing the cells to properly sense and respond to the surrounding mechanical microenvironments (*Science* 2009;324:1673-7; *Nature Methods* 2018;15:491-8).

In the revised manuscript, we isolated soft MCF-7 cells by using the microfluidic chip and then seeded these soft cells in soft 3D fibrin gel for 4, 6, 12, 24, 48, 72 or 96 hr. The cellular

stiffness of these cells was detected by AFM. The result showed that during the observation period, these soft MCF-7 cells always kept their mechanical character of being softness, suggesting that stiffness is an intrinsic character of a cell.

According to the reviewer's suggestion, we added this result in the revised manuscript, page 7 line 20 and Fig EV2B.

5. *There appears to be some conceptual overlap from earlier studies involving some of these authors, e.g. Tan et al Nat Comm 5: 4619 (2014). The authors should incorporate this paper more explicitly in their introduction and explain more clearly the new advances here.*

Response:

According to the reviewer's suggestion, we added this information in the section of introduction in the revised manuscript, page 3 line 20.

"This suggests that soft cells should survive in a soft stroma and stiff cells function optimally in a stiff niche. As a support, we have demonstrated that 3D soft fibrin matrices promote H3K9 demethylation and increase Sox2 expression and self-renewal of melanoma stem cells, whereas stiff ones exert opposite effects (*Nat Commun.* 2014;5:4619). In line with this notion, cells with various degrees of stiffness can co-exist within the same tumor tissue, due to the heterogeneity of the tumor mechanical microenvironments (*Nat Nanotechnol.* 2012;7:757-65; *Nat Mater.* 2014;13:631-7). Despite such understanding, direct evidence that the cellular softness functions as a basic feature for tumorigenic cells remains elusive. In this study, we develop a method to separate soft cells from stiff ones and provide evidence that these soft cells are highly tumorigenic and possess the ability to metastasize".

6. *Based on publicly available tumor sequence databases (e.g. TCGA), is BCL9L expression associated with malignancy and/or poor prognosis?*

Response:

We thank the reviewer's comment. In our original manuscript, we indeed investigated whether BCL9L expression is associated with patients' prognosis, and found that the higher expression of BCL9L in tumor tissues was correlated with a poorer prognosis of patients with melanoma, breast cancer and other cancer types such as pancreatic cancer, lung cancer and liver cancer (please see original Fig 6F and EV5J).

RESPONSE TO REVIEWER #2:

Referee #2:

In this study, the investigators explore the potential of cellular softness as a marker of tumor-initiating and metastatic cancer cells. They use a specific microfluidic chip to isolate stiff and soft cancer cells from different malignancies, and this is followed by characterization of the isolated cells in vitro and in vivo. The results indicate that soft cancer cells have tumor initiating capacity and are more metastatic than stiff cells. Interestingly, this is independent of previously identified cancer stem cell (CSC) markers. The authors use transcriptomic analysis to identify BCL9L, a co-factor of beta-catenin and regulator of Wnt signaling, as a functional mediator of aggressivity in soft cancer cells. Finally, the investigators analyze BCL9L expression in clinical cancer samples and identify association between BCL9L and poor patient outcome. This is a very interesting and overall thorough study. However, I do have some critical points that need to be addressed. Details below.

1. The investigators characterize soft cancer cells and demonstrate that this property is an excellent marker of tumor-initiating ability and cancer aggressiveness across different cancer types. Growing evidence indicates that tumor-initiating properties can be surprisingly plastic and context dependent, suggesting that cancer cells may gain or lose these properties depending on the microenvironment. The authors state that physical traits may be more stable than expression of CSC marker genes. With this in mind, I think it would be important to show that different levels of stiffness are indeed stable properties. What happens when selected soft cells are grown for some time in culture? Do they maintain the soft phenotype? What about growth of selected cells in stiff matrix rather than soft matrix? Does it influence the cellular stiffness.

Response:

We thank the reviewer's pertinent comments on cancer cells gaining or losing tumor-initiating properties depending on the microenvironment. Stiffness is an inherent feature of a cell, which is mainly provided by F-actin filaments (*Science* 1993;260:1124-7). Different cell types display varying levels of stiffness, which matches the stiffness of the local extracellular matrix, allowing the cells to properly sense and respond to the surrounding mechanical microenvironments (*Science* 2009;324:1673-7; *Nature Methods* 2018;15:491-8).

In the revised manuscript, we isolated soft MCF-7 cells by using the microfluidic chip and cultured the cells in soft 3D fibrin gel or stiff flask for 4, 6, 12, 24, 48, 72 and 96 hr. Stiffness of the cells was determined by atomic force microscopy (AFM). We found that the softness of the isolated soft MCF-7 cells could be kept in soft fibrin gels, which, however, started to become stiff following 4-hour culture in stiff flask, and reached the peak 12 hours later.

According to the reviewer's suggestion, we added this result in the revised manuscript, page 7 line 20 and Fig EV2B and C.

2. Following up on the previous point, is it fair to use soft matrix when colony formation is measured to compare soft and stiff cells? How would the comparison turn out in stiff matrix? Would this now benefit stiff cells?

Response:

Seeding the soft MCF-7 cells isolated by the microfluidic chip in 90 Pa soft 3D fibrin gels could keep the softness of the cells, however, seeding soft tumor cells into 1050 Pa stiff 3D fibrin gels induced cells into dormancy (*Cancer Res.* 2018;78(14):3926-3937). In contrast to soft MCF-7 cells, if seeding the stiff MCF-7 cells into 90 Pa soft 3D fibrin gels, the formed colony number decreased dramatically and displayed a differentiated morphology (please see original Figure 2A); however, such stiff cells grew well in rigid plate. Thus, only the soft cells can grow colonies in soft fibrin gels but stiff ones cannot. This means that we can use colony formation in soft 3D fibrin gels to further verify the softness of the isolated soft tumor cells through microfluidic chip.

3. In Figure 6B and 6E, the authors address correlation between tumor grade and the percentage of soft cells or the levels of BCL9L in patient samples. However, the graphs presented are not appropriate to address this. A simple correlation plot with the parameters on x and y axis would be more appropriate. For statistical analysis, the investigators use Pearson's correlation test and thus the correlation coefficient R should be included.

Response:

We thank the reviewer's comment. Accordingly, we presented the graphs for Figure 6B and 6E with the "Soft cells (%)" or "Relative BCL9L level" on x axis and "Grade of cancer" on y axis, and also included the correlation coefficient R in the revised manuscript.

4. It is interesting to see how much better of a marker the cellular softness can be compared to previously recognized CSC markers (Figure 3 and EV3). Notably, a number of the CSC markers (ALDH, Side Population and CD133) were significantly associated with increased softness. Moreover, the combination of ALDH and softness, as markers to isolate cancer cells, modestly improved tumor initiation in mice compared to softness alone (Fig 3B). Not all soft cells express CSC markers and not all CSC marker positive cells are soft. Therefore, it would be insightful to explore the combination of softness and CSC markers a bit further. Does the combination of softness and other CSC markers select for more enhanced tumor-initiation or metastatic ability compared to softness alone. Further analysis of the double positive cells (soft + CSC markers) is warranted.

Response:

We appreciate the reviewer's comment. In our unpublished data, we had sorted the soft cells from bulk 4T1 and ALDH⁺ 4T1 cells, respectively, by using microfluidic chip. Then, we injected these different soft cells into the mammary fat pads of NSG mice (100 cells/mouse). Twelve weeks later, the tumor formation was recorded. The result showed that the soft 4T1 cells and the soft ALDH⁺ 4T1 cells had the similar tumorigenic capability (5/10 versus 6/10). Thus, the intrinsic softness is a suitable characteristic to mark tumorigenic cells.

According to the reviewer's concern, we added this result in the revised manuscript, page 9 line 17 and Fig EV3B.

5. The results presented in this manuscript should be discussed further in the context of the large amount of studies showing positive association between increased tissue stiffness and aggressive cancer behavior. Here the authors show that soft cells are more mobile and invasive. However, substantial evidence suggests that cross-linked collagen mediated by LOX family members and leading to increased stiffness is associated with increased invasion and aggressive cancer behavior.

Response:

We thank the reviewer's constructive comment. Accordingly, we discussed the positive association between increased tissue stiffness and aggressive cancer behavior as below:

“Published reports have shown a positive association between increased tissue stiffness and aggressive cancer behavior, prompting to propose a model of cancer progression that depends on static or dynamic tumor tissue stiffening (*Proc Natl Acad Sci. 2019;116:3502-3507; Cancer cell. 2005;8:241-254; Nat Cell Biol. 2015;17:678-688*). However, there has been no evidence to support the notion that the cells coming out of the stiffened tumor stroma are tumorigenic and metastatic cancer cells. It has also been unclear whether the tumorigenic cells are stiff cells or soft cells. Despite the overall stiffness, local microenvironments for tumor stiffness are highly heterogeneous (*Nat Nanotechnol. 2012;7:757–65*). Increased tissue stiffness may be attributed to more extracellular matrices, which are likely to limit blood vessel distribution and lead to tumor hypoxia, a common phenomenon in tumor microenvironments. It is known that hypoxia in primary tumors is associated with an increased metastasis and a worse prognosis in cancer patients (*Nature. 2006;440:1222-6; Science. 2016;352:175-80; Nat Rev Cancer. 2014;14:430-9*). Recently, we demonstrated that hypoxia promote human breast tumor-repopulating cell development (*Oncogene. 2019;38:6970-6984*). Of note, hypoxic areas may be very soft due to local tissue necrosis and matrix degradation. Thus, increased tissue stiffness may result in more soft tumor cells at the hypoxic sites, favoring an aggressive cancer behavior. Consistently, studies by Superfine et al showed that cancer cells with the highest migratory and invasive potential are five times less stiff than cells with the lowest migration and invasion potential (*Cancer Res. 2011;71:5075–80*)”.

According to the reviewer's suggestion, we added this information in the revised manuscript, page 15 line 8.

6. It is not clear to this reviewer what is being compared in Figure 6D.

Response:

We thank the reviewer's indicating this detail. In original Figure 6D, we stained the tissue sections from 6 high-grade melanoma of patients with anti-BCL9L antibody. The result showed that BCL9L was highly expressed in melanoma tissues. As a comparison, 3 patients' low-grade melanoma were stained and the result showed a decreased expression of BCL9L.

According to the reviewer's concern, we added the result from the 3 low-grade melanoma samples in the revised manuscript, revised Fig 6D.

7. *For BCL9L knockout, CRISPR.EFS.GFP positive cells were selected by flow cytometry based on GFP expression. However, if guide RNA against GFP was used as a control, would this not affect selection of CRISPR.EFS.GFP positive cells?*

Response:

We thank the reviewer's indicating this error. In the revised manuscript, we corrected the original sentence with "These SGRNAs were cloned into the pL-CRISPR.EFS.RFP vector plasmid (addgene, #57819)" and replaced "GFP-positive cells were sorted by flow cytometry" with "RFP-positive cells were sorted by flow cytometry".

8. *It is not clear how the metastasis index is derived. What does it mean specifically?*

Response:

We thank the reviewer's indicating this detail. The metastasis index was defined as the percentage of metastasis tumor areas to total lung areas based on calculation from 10 slides, according to the protocol we published before (*Cancer Immunol Res.* 2018;6:1046-1056).

According to the reviewer's concern, we added this information in the revised manuscript, page 37 line 2.

9. *There is a discordance between text and Figure 2E legend. The figure legend states "(E) the same as (B)", essentially primary tumor analysis. However, the main text states that metastatic colonization of the lungs was analyzed.*

Response:

We thank the reviewer's indicating this error. In the revised manuscript, we revised the figure legend for Figure 2E as "Stiff or soft B16 or MP-1 cells (100 or 10 cells) were injected into the NSG mice by tail vein".

10. *Total beta-catenin expression is not a good marker of active Wnt signaling. Nuclear beta-catenin or a reporter are more reliable.*

Response:

We thank the reviewer's constructive comments. Accordingly, we isolated the total nuclear proteins from stiff or soft 4T1, MCF-7, B16 or MP-1 cells and determined the expression of β -catenin by western blot. The result showed that the expression of nuclear β -catenin was increased in the soft tumor cells, which was consistent with the result from immunostaining of β -catenin (please see original Figure EV5C).

According to the reviewer's concern, we added this result in the revised manuscript, page

12 line 8 and Fig EV5D.

Thank you for submitting a revised version of your manuscript. Your study has now been seen by both original referees, who find that their main concerns have been addressed and support publication of the revised manuscript. There now remain only a few mainly editorial issues that have to be addressed before I can extend formal acceptance of the manuscript:

1. Please address the remaining minor point from reviewer #1.

Please let me know if you have any further questions regarding any of these points. You can use the link below to upload the revised files.

Referee #1:

In general, the authors have responded well to my comments. My only remaining comment has to do with their response to my comment #4 and reviewer 2's comment #1 regarding the stability of cell softness values. The data in EV2B-C nicely address the concern, but the associated text is unclear as written and should be revised ("Meanwhile, we found that the softness of the isolated soft MCF-7 cells could be kept in soft fibrin gels, which, however, started to become stiff following 4-hour culture in conventional two-dimensional (2D) rigid dishes, and reached the peak 12 hours later (Fig EV2B 3 and C)"). That minor concern aside, the work seems ready for publication.

Referee #2:

The investigators have made significant revisions to the manuscript and my critical points have been thoroughly addressed. I do not have further comments on this interesting study.

RESPONSES TO THE EDITOR AND REVIEWERS

We would like to express our sincere thanks to you and both reviewers for their critical and constructive comments. We respond point-by-point to their comments. We feel that their comments have helped us on improving and strengthening the manuscript.

RESPONSE TO THE EDITOR

1. Please address the remaining minor point from reviewer #1.

Response:

We appreciate the reviewer's comment. Accordingly, in the revised manuscript, we revised the sentence as below:

Meanwhile, we found that the isolated soft MCF-7 cells could maintain their softness in the soft fibrin gels, while, the soft cells, if seeded in rigid culture plates for 4 hours, started to become stiff and reached the stiff peak 12-hour later (Fig EV2B and C).

Editor accepted the manuscript.

Corresponding Author Name: Yuying Liu and Bo Huang

Journal Submitted to: The EMBO Journal

Manuscript Number: EMBOJ-2020-106123R